# Genome-wide analysis of R2R3-MYB transcription factors in Japanese morning glory

Ayane Komatsuzaki[1], Atsushi Hoshino[2,3], Shungo Otagaki[1], Shogo Matsumoto[1], Katsuhiro Shiratake[1]*

**1** Graduate School of Bioagricultural Sciences, Nagoya University, Nagoya, Japan, **2** National Institute for Basic Biology, Okazaki, Japan, **3** Department of Basic Biology, SOKENDAI (The Graduate University for Advanced Studies), Okazaki, Japan

* shira@agr.nagoya-u.ac.jp

**Data Availability Statement:** All relevant data are within the paper and its Supporting Information files.

## Abstract

The R2R3-MYB transcription factor is one of the largest transcription factor families in plants. R2R3-MYBs play a variety of functions in plants, such as cell fate determination, organ and tissue differentiations, primary and secondary metabolisms, stress and defense responses and other physiological processes. The Japanese morning glory (*Ipomoea nil*) has been widely used as a model plant for flowering and morphological studies. In the present study, 127 *R2R3-MYB* genes were identified in the Japanese morning glory genome. Information, including gene structure, protein motif, chromosomal location and gene expression, were assigned to the *InR2R3-MYBs*. Phylogenetic tree analysis revealed that the 127 InR2R3-MYBs were classified into 29 subfamilies (C1-C29). Herein, physiological functions of the InR2R3-MYBs are discussed based on the functions of their *Arabidopsis* orthologues. InR2R3-MYBs in C9, C15, C16 or C28 may regulate cell division, flavonol biosynthesis, anthocyanin biosynthesis or response to abiotic stress, respectively. C16 harbors the known anthocyanin biosynthesis regulator, InMYB1 (INIL00g10723), and putative anthocyanin biosynthesis regulators, InMYB2 (INIL05g09650) and InMYB3 (INIL05g09651). In addition, INIL05g09649, INIL11g40874 and INIL11g40875 in C16 were suggested as novel anthocyanin biosynthesis regulators. We organized the *R2R3-MYB* transcription factors in the morning glory genome and assigned information to gene and protein structures and presuming their functions. Our study is expected to facilitate future research on *R2R3-MYB* transcription factors in Japanese morning glory.

## Introduction

Transcription factors (TFs) are essential for the regulation of gene expression. Specific binding of TFs to cis-elements in promoter regions of genes activates or represses gene expression, thereby controlling various physiological events, such as tissue and organ developments, metabolic processes, and stress responses [1–4]. A large number of TF genes are present in the

**Funding:** This work was supported partially by the Grant-in-Aids for Scientific Research (KAKENHI: 15H04449, 16K14850, 20K20372, 18H03950, 21K19111, 21H02184) to K.S. from the Japan Society for the Promotion of Science (JSPS: https://www.jsps.go.jp/english/index.html). The funders had no role in study design, data collection and analysis, decision to publish, or preparation of the manuscript.

**Competing interests:** The authors have declared that no competing interests exist.

plant genome, accounting for about 7% of all genes in the genome [5]. TFs can be classified into different families according to the conserved amino acid sequences in their DNA binding domains [6].

MYB TF family is among the largest TF families in plants and possess a highly conserved MYB DNA-binding domain consisting of 1–4 imperfect MYB repeat sequences on the N-terminal side [2]. Each repeat consists of 50–55 amino acid residues, containing three regularly spaced tryptophan (W) residues that together form a helix-turn-helix hold. The second and third α-helices in each repeat interacts with the major DNA groove [7, 8]. MYB proteins can be classified into four subfamilies according to the number of MYB repeats: 1R-MYB, 2R (R2R3)-MYB, 3R(R1R2R3)-MYB, and 4R-MYB, harboring one, two, three, or four MYB repeats, respectively [9, 10]. 1R-MYBs are also called MYB-related proteins. R2R3-MYBs are dominant in plants [2].

After the identification of the first MYB TF in plants, various MYB TF family members have been identified and characterized in a wide range of plant species, including *Arabidopsis thaliana* [9], *Solanum lycopersicum* [11], *Poplus trichocarpa* [12], *Zea mays* [13], *Glycine max* [14], *Malus domestica* [15], *Beta vulgaris* [16], and *Solanum tuberosum* [17]. The functions of MYBs, particularly that of R2R3-MYBs, have been thoroughly investigated. R2R3-MYBs play important roles in various biological processes, including regulation of cell cycle, cell fate and differentiation, tissue and organ development, primary and secondary metabolism, hormone biosynthesis and signal transduction, and response to biotic and abiotic stress [9]. For example, AtMYB75 (PAP1), AtMYB90 (PAP2), AtMYB113 and AtMYB114 in *Arabidopsis* [18, 19], AN2 in petunia [20], MdMYBA, MdMYB1, MdMYB10 and MdMYB110a in apple [21, 22], SlMYB12 in tomato [23], and StAN1 in potato [11, 24] regulate anthocyanin biosynthesis. MIXTA, an R2R3-MYB in *Antirrhinum majus*, was reported to regulate elongation of epidermal cells in petals and leaves, as well as trichome formation [25, 26]. MIXTA-like proteins, including AmMYBML1, AmMYBML2 and AmMYBML3 of *A. majus*, PhMYB1 of petunia, and AtMYB16 and AtMYB106 of *Arabidopsis*, are known to have similar functions as the MIXTA [26–29].

Japanese morning glory (*Ipomoea nil*) has been a traditional floricultural plant in Japan since the 17th century. It is widely used as a model plant for studies of flowering, flower color, and floral organ morphology because of its keen day-length sensitivity and the existence of various varieties and mutants in flower color and morphology [30–34]. Notably, the whole genome of this plant has been sequenced [35]. The genome information and bio-resources of morning glory, including information on seeds of more than 1,500 cultivars and mutants and genomic and cDNA clones, are available from the National BioResource Project of the Ministry of Education, Culture, Sports, Science and Technology (https://shigen.nig.ac.jp/asagao/).

The major mutagens in Japanese morning glory are *Tpn1* family transposons [30] and most flower color mutations are caused by transposon insertion into anthocyanin biosynthesis genes, such as *chalcone synthase*, *chalcone isomerase*, *dihydroflavonol 4-reductase*, and *UDP-glucose:flavonoid 3-O-glucosyltransferase* [36–39].

These anthocyanin biosynthesis genes are regulated by R2R3-MYB TF, bHLH (basic-Helix-Loop-Helix) protein, and WD repeat (tryptophan-aspartic acid repeat, WDR) protein [40, 41]. In Japanese morning glory, InMYB1 and InWDR1 were identified as regulators of anthocyanin biosynthesis [42], and the insertion of the Stowaway-like transposon *InSto1* into the *InWDR1* gene causes flower color mutation [43]. In addition, mutations in TFs causes changes in the morphology of morning glory flowers. For example, double-flower mutation is caused by transposon insertion in the MADS-box TF [44], and separated or tubular petal mutation is caused by a transposon insertion into the *GARP* TF [45]. The functions of TFs in determining the color and morphology of flowers have been studied not only in morning glory but also in

other floricultural plants, and TFs have been used as targets in molecular breeding to change flower color and morphology (http://www.cres-t.org/fiore/public_db/index.shtml) [46].

As described above, R2R3-MYBs play various important roles in plants, however, information on R2R3-MYBs in morning glory is currently limited. Clarification of R2R3-MYB functions in this plant is important for understanding of flowering, coloration and morphology, and other important traits of flowers. The Japanese morning glory 'Tokyo Kokei Standard line' genome (750 Mb) has been sequenced up to 98%, and scaffolds covering 91.42% of the assembly have been anchored to 15 pseudo-chromosomes [35]. In this study, we identified 126 genes encoding R2R3-MYB TFs in the Japanese morning glory genome and assigned and listed their information, such as gene ID, gene structure, protein motif, chromosomal location, gene expression profile, and physiological functions.

## Materials and methods

### Identification of R2R3-MYBs in Japanese morning glory genome

To identify candidates of Japanese morning glory R2R3-MYBs, we performed a BLASTP search (e-value < 1e-10) of the Japanese morning glory genome database (http://viewer. shigen.info/asagao/) [35] using the Hidden Markov Model profile of the MYB binding domain (PF00249) from the Pfam (http://pfam.xfam.org/) [47] and the amino acid sequences of *Arabidopsis* R2R3-MYBs [9] as queries. Against the obtained non-redundant candidates of R2R3-MYBs, presence of the conserved MYB domain was confirmed by Pfam, SMART (http://smart.embl-heidelberg.de/) [48, 49] and PROSITE (https://prosite.expasy.org/) [50]. Candidates harboring two MYB domains (R2 and R3 domains) predicted by all Pfam, SMART and PROSITE were identified as Japanese morning glory R2R3-MYBs.

### Multiple sequence alignment and phylogenic analysis

The amino acid sequences of R2R3-MYBs were aligned using the ClustalW program [51], and an unrooted neighbor-joining phylogenetic tree was constructed using MEGA X [52] with the following parameters: Poisson model, pairwise deletion and 1,000 bootstrap replications. The Japanese morning glory R2R3-MYBs were classified based on a boostrap value of 50 or higher. However, even if the boostrap value was below 50, R2R3-MYBs associated with a particular subgroup of Arabidopsis R2R3-MYB were treated as a single clade. R2R3-MYBs that did not meet this condition were not considered to belong to any clade.

### Gene structure and protein motif analyses

Gene structure (exon-intron structure) was schematized with the coding sequences and the genomic sequence of the Japanese morning glory *R2R3-MYB*s by the Gene Structure Display Server (GSDS: http://gsds.gao-lab.org/) [53]. Multiple Expectation Maximization for Motif Elicitation (MEME: https://meme-suite.org/meme/tools/meme) [54] was used to identify the conserved protein motifs of the R2R3-MYBs, with the following parameters: the maximum number of motifs was set to identify 20 motifs and optimum width of motifs was set from six to 100 amino acids.

### Chromosomal location and gene duplication analysis

Information on the chromosome distribution of the *InR2R3-MYB* genes was obtained from the Japanese morning glory genome database, while MapChart [55] was used for the graphical presentation of chromosomal location. Tandemly duplicated genes were defined as an array of two or more *InR2R3-MYB* genes falling within 100 kb of one another.

### In silico gene expression analysis

Gene expression data for various organs of morning glory were downloaded from the Japanese morning glory genome database (http://viewer.shigen.info/asagao/jbrowse.php?data=data/Asagao_1.2/) and converted to 10 logarithms. A heatmap was then created using R package gplots (https://cran.r-project.org/web/packages/gplots/index.html). Details of the samples were described in Hoshino et al., (2016) [35]. Briefly, the embryo is immature green embryos; the flower includes fully opened flowers and flower buds at various stages; the leaf includes leaves of various sizes; the stem includes young stems with shoot tips; the seed includes seed coats at various developmental stages; the root is three-weeks-old roots.

## Results and discussion

### Identification and classification of the morning glory R2R3-MYBs

To identify the morning glory R2R3-MYBs, we performed a BLAST search of the morning glory genome database (http://viewer.shigen.info/asagao/) using the MYB domain (PF00249) from Pfam and 126 *Arabidopsis* R2R3-MYBs [9] as queries. A total of 270 candidates were identified from the BLAST search. Subsequently, the presence of the MYB domains was confirmed by the Pfam, SMART and PROSITE. As a result, 126 R2R3-MYBs, harboring two MYB domains, were identified (Table 1 and S1 Table). Among them, three InR2R3-MYBs, InMYB1, InMYB2 and InMYB3, had been reported [42]. Although INIL05g09651, which has the highest homology to the InMYB3, harbors only one MYB domain, INIL05g09651 was considered to be identical with InMYB3, thus, it was included in R2R3-MYBs. Finally, total 127 R2R3-MYBs were identified in the morning glory genome. Forty 1R-MYBs (MYB-related proteins), three R1R2R3-MYBs (3R-MYBs) and one 4R-MYB harboring one, three or four MYB domains, respectively, were also listed in S2 Table.

 *Arabidopsis* R2R3-MYBs were classified into 23 functional subgroups (S1-S25) [9], some of which have been well characterized. To understand the evolutionary relationship between the R2R3-MYBs of morning glory and *Arabidopsis* and predict the functions of the morning glory R2R3-MYBs using those of *Arabidopsis* orthologues, a phylogenetic tree of the R2R3-MYBs was constructed (Fig 1). The phylogenetic trees revealed 29 subfamilies (C1-C29) of the morning glory R2R3-MYBs (InR2R3-MYBs). Eleven InR2R3-MYBs did not belong to any clade, while subfamily S12 of *Arabidopsis* R2R3-MYBs was absent in morning glory. *Arabidopsis* R2R3-MYBs belonging to S12 have been reported to regulate glucosinolate biosynthesis [56–58]. Glucosinolates are unique secondary metabolites in *Brassicaceae* [57], therefore morning glory has no homolog of Arabidopsis R2R3-MYBs in S12. On the other hand, C7 and C19 were found to be unique subfamilies in morning glory, suggesting that they might be responsible for unique functions in morning glory.

### Gene structure and protein motif of InR2R3-MYBs

Gene structure (exon-intron structure) suggests the evolutionary background of genes. The gene structures of *InR2R3-MYBs* are shown in Fig 2B. *INIL02g16845* in C6 and *INIL08g38640* and *INIL15g23810* in C25 have a large number of exons, i.e. 13 exons or 12 exons, respectively. In contrast, *INIL01g25379*, *INIL01g25431*, *INIL06g38304*, *INIL14g41452*, *INIL14g41510* and *INIL15g27998* in C28 and *INIL04g32702* belonging to no clade have only one exon each.

 Conserved protein motifs among the InR2R3-MYBs were determined using MEME (https://meme-suite.org/meme/tools/meme), and 20 conserved motifs were identified (Fig 2C and S3 Table). Most of InR2R3-MYBs has five highly conserved motifs in the same order; motif 3, 5/13, 1, 4 and 2. Motif 3 or motif 5/13 comprise helices 1 and 2 in the R2 repeat,

**Table 1. The list of the R2R3-MYBs identified in the genome of Japanese morning glory.**

| Subfamily | | Gene ID | Gnomon (NCBI) | Chromosome number | Position on chromosome | | strand | Number of amino acids | Number of exons |
|---|---|---|---|---|---|---|---|---|---|
| Morning glory | Arabidopsis | | | | start | end | | | |
| C1 | S24 | INIL02g11823 | XM_019329763.1 | 2 | 2,358,987 | 2,360,659 | + | 317 | 4 |
| C1 | S24 | INIL11g16076 | XM_019334837.1 | 11 | 14,828,803 | 14,831,927 | + | 307 | 3 |
| C2 | | INIL04g04429 | XM_019320704.1 XM_019320705.1 XM_019320706.1 XM_019320707.1 | 4 | 20,917,488 | 20,922,002 | - | 318 | 3 |
| C2 | | INIL12g01471 | XM_019334603.1 | 12 | 898,729 | 901,810 | + | 309 | 3 |
| C3 | S11 | INIL03g17749 | XM_019337779.1 | 3 | 34,071,898 | 34,073,432 | + | 353 | 3 |
| C3 | S11 | INIL04g32440 | XM_019306900.1 | 4 | 3,676,755 | 3,678,276 | - | 378 | 3 |
| C3 | S11 | INIL09g30441 | XM_019303428.1 | 9 | 2,892,833 | 2,894,445 | + | 352 | 3 |
| C3 | S11 | INIL14g04070 | XM_019320491.1 | 14 | 5,046,365 | 5,047,724 | - | 326 | 3 |
| C4 | S1 | INIL03g21132 | XM_019342517.1 | 3 | 24,857,256 | 24,859,767 | - | 321 | 3 |
| C4 | S1 | INIL05g28446 | XM_019301469.1 | 5 | 5,753,292 | 5,754,886 | - | 308 | 3 |
| C4 | S1 | INIL09g35855 | XM_019311436.1 | 9 | 13,796,939 | 13,798,876 | - | 313 | 3 |
| C4 | S1 | INIL10g12662 | XM_019332084.1 | 10 | 4,262,363 | 4,264,568 | + | 311 | 3 |
| C5 | S9 | INIL08g38600 | XM_019315091.1 | 8 | 38,488,785 | 38,490,624 | - | 336 | 3 |
| C5 | S9 | INIL08g38603 | XM_019315089.1 | 8 | 38,538,443 | 38,543,065 | + | 336 | 3 |
| C5 | S9 | INIL08g38605 | XM_019314993.1 | 8 | 38,602,546 | 38,607,973 | + | 281 | 4 |
| C5 | S9 | INIL08g38606 | XM_019314995.1 | 8 | 38,655,992 | 38,666,483 | - | 333 | 3 |
| C5 | S9 | INIL08g38607 | XM_019315080.1 | 8 | 38,736,149 | 38,738,540 | - | 333 | 3 |
| C5 | S9 | INIL11g18710 | XM_019339411.1 | 11 | 8,223,605 | 8,225,759 | + | 392 | 3 |
| C6 | S2 | INIL12g08387 | XM_019325800.1 | 12 | 7,229,877 | 7,231,502 | + | 249 | 3 |
| C6 | S2 | INIL13g08188 | XM_019324572.1 XM_019324573.1 XM_019324574.1 | 13 | 1,602,872 | 1,604,928 | - | 261 | 3 |
| C7 | | INIL06g15152 | XM_019334487.1 | 6 | 47,132,344 | 47,133,424 | + | 237 | 3 |
| C7 | | INIL06g15156 | XM_019334410.1 | 6 | 47,077,003 | 47,078,600 | + | 267 | 3 |
| C7 | | INIL09g30444 | XM_019303240.1 | 9 | 2,937,097 | 2,938,314 | - | 266 | 3 |
| C7 | | INIL09g30445 | XM_019303239.1 | 9 | 2,940,343 | 2,941,614 | - | 266 | 3 |
| C7 | | INIL09g30446 | XM_019303432.1 | 9 | 2,952,676 | 2,953,774 | - | 260 | 3 |
| C8 | | INIL01g00015 | XM_019323694.1 | 1 | 7,267,332 | 7,269,516 | + | 310 | 2 |
| C8 | | INIL02g11914 | XM_019330080.1 | 2 | 3,350,345 | 3,351,718 | - | 296 | 4 |
| C8 | | INIL02g17162 | XM_019336461.1 | 2 | 42,689,808 | 42,691,054 | + | 269 | 2 |
| C8 | | INIL05g09388 | XM_019327136.1 | 5 | 1,193,059 | 1,194,487 | - | 283 | 3 |
| C8 | | INIL08g04788 | XM_019321152.1 | 8 | 3,236,853 | 3,239,338 | + | 240 | 3 |
| C8 | | INIL15g31180 | XM_019304955.1 | 15 | 9,663,646 | 9,667,201 | + | 261 | 2 |
| C9 | S14 | INIL01g36710 | XM_019312234.1 | 1 | 5,231,116 | 5,238,495 | + | 273 | 4 |
| C9 | S14 | INIL02g11599 | XM_019330240.1 | 2 | 753,230 | 754,530 | - | 334 | 3 |
| C9 | S14 | INIL02g16681 | XM_019336497.1 | 2 | 39,162,404 | 39,163,938 | + | 277 | 3 |
| C9 | S14 | INIL05g09674 | XM_019326908.1 | 5 | 3,532,267 | 3,538,665 | + | 364 | 4 |
| C9 | S14 | INIL08g00165 | XM_019333004.1 | 8 | 27,155,955 | 27,168,867 | + | 399 | 6 |
| C9 | S14 | INIL08g30969 | XM_019304699.1 | 8 | 7,648,225 | 7,650,399 | - | 334 | 3 |
| C9 | S14 | INIL10g12144 | XM_019331575.1 | 10 | 286,800 | 288,308 | - | 325 | 3 |
| C9 | S14 | INIL11g10021 | XM_019327588.1 | 11 | 507,292 | 508,821 | - | 312 | 4 |
| C9 | S14 | INIL11g18940 | XM_019340046.1 | 11 | 6,203,502 | 6,204,856 | + | 256 | 3 |
| C9 | S14 | INIL12g22053 | XM_019343471.1 | 12 | 59,731,879 | 59,733,712 | + | 326 | 4 |
| C9 | S14 | INIL14g41566 | XM_019318620.1 | 14 | 57,516,461 | 57,524,802 | + | 377 | 3 |
| C10 | S16 | INIL05g22908 | XM_019344838.1 | 5 | 32,521,998 | 32,526,737 | - | 289 | 3 |

*(Continued)*

**Table 1.** (Continued)

| Subfamily | | Gene ID | Gnomon (NCBI) | Chromosome number | Position on chromosome | | strand | Number of amino acids | Number of exons |
|---|---|---|---|---|---|---|---|---|---|
| Morning glory | Arabidopsis | | | | start | end | | | |
| C10 | S16 | INIL11g18972 | XM_019339450.1 | 11 | 5,950,686 | 5,952,942 | - | 294 | 3 |
| C11 | S13 | INIL10g16278 | XM_019335486.1 | 10 | 15,049,362 | 15,051,304 | - | 330 | 5 |
| C11 | S13 | INIL14g06864 | XM_019323598.1 | 14 | 39,922,743 | 39,924,269 | + | 273 | 3 |
| C12 | | INIL03g11384 | XM_019329572.1 | 3 | 28,505,899 | 28,520,498 | + | 499 | 6 |
| C12 | | INIL09g26302 | XM_019298694.1 | 9 | 36,703,389 | 36,705,523 | + | 319 | 3 |
| C13 | | INIL03g18278 | XM_019339022.1 | 3 | 37,886,710 | 37,892,779 | + | 354 | 3 |
| C13 | | INIL05g24002 | XM_019295562.1 | 5 | 36,357,513 | 36,361,037 | - | 311 | 3 |
| C13 | | INIL06g23639 | XM_019295011.1 XM_019295013.1 | 6 | 39,073,755 | 39,080,094 | + | 315 | 3 |
| C13 | | INIL06g37657 | XM_019313132.1 XM_019313133.1 | 6 | 6,003,732 | 6,005,207 | + | 330 | 2 |
| C14 | S4 | INIL02g10399 | XM_019328601.1 | 2 | 23,136,031 | 23,137,735 | - | 204 | 3 |
| C14 | S4 | INIL04g34740 | XM_019309686.1 XM_019309687.1 | 4 | 337,406 | 338,761 | + | 286 | 2 |
| C14 | S4 | INIL05g22708 | XM_019345006.1 | 5 | 30,885,896 | 30,887,485 | - | 298 | 3 |
| C14 | S4 | INIL08g31096 | XM_019304555.1 | 8 | 6,205,257 | 6,209,143 | - | 177 | 3 |
| C14 | S4 | INIL11g10884 | XM_019328930.1 | 11 | 37,028,417 | 37,030,121 | - | 271 | 3 |
| C14 | S4 | INIL14g35341 | XM_019310094.1 | 14 | 50,845,779 | 50,846,985 | - | 253 | 2 |
| C15 | S7 | INIL08g13530 | XM_019332820.1 XM_019332821.1 | 8 | 8,429,287 | 8,435,837 | - | 377 | 3 |
| C15 | S7 | INIL13g07908 | XM_019324780.1 | 13 | 3,578,882 | 3,581,573 | - | 338 | 3 |
| C16 | S6 | INIL05g09649 | XM_019327061.1 XM_019327062.1 | 5 | 3,227,339 | 3,229,379 | - | 264 | 3 |
| C16 | S6 | INIL05g09650 (InMYB2) | XM_019327060.1 | 5 | 3,269,350 | 3,271,009 | - | 261 | 3 |
| C16 | S6 | INIL00g10723 (InMYB1) | XM_019328791.1 | scaffold0894 | 11,156 | 19,420 | + | 370 | 4 |
| C16 | S6 | INIL11g40874 | - | 11 | 18,384,802 | 18,392,696 | - | 167 | 4 |
| C16 | S6 | INIL11g40875 | XM_019317839.1 | 11 | 18,354,162 | 18,363,866 | - | 509 | 6 |
| C16 | S6 | "INIL05g09651 (InMYB3)" | XM_019327050.1 | 5 | 3,308,001 | 3,309,889 | - | 221 | 3 |
| C17 | S15 | INIL02g10645 | XM_019328367.1 | 2 | 36,405,994 | 36,407,651 | - | 220 | 3 |
| C17 | S15 | INIL13g40955 | XM_019317956.1 XM_019317957.1 | 13 | 18,175,994 | 18,177,378 | + | 187 | 3 |
| C18 | | INIL12g24707 | XM_019296345.1 | 12 | 49,224,678 | 49,234,268 | + | 281 | 3 |
| C18 | | INIL12g24714 | XM_019296352.1 | 12 | 49,052,482 | 49,081,722 | + | 465 | 5 |
| C18 | | INIL00g14902 | XM_019334094.1 | scaffold1249 | 12,847 | 14,357 | - | 306 | 3 |
| C19 | | INIL05g23059 | XM_019344607.1 XM_019344608.1 | 5 | 33,970,297 | 33,971,627 | - | 277 | 2 |
| C19 | | INIL06g37604 | XM_019313528.1 | 6 | 6,532,453 | 6,538,389 | + | 261 | 3 |
| C19 | | INIL06g37606 | XM_019313505.1 | 6 | 6,503,384 | 6,505,212 | + | 283 | 2 |
| C19 | | INIL15g31267 | XM_019304856.1 | 15 | 8,040,255 | 8,042,080 | + | 316 | 2 |
| C20 | | INIL05g04549 | XM_019320800.1 | 5 | 4,950,049 | 4,951,766 | - | 245 | 3 |
| C20 | | INIL13g08245 | XM_019325222.1 | 13 | 1,195,101 | 1,196,553 | - | 245 | 3 |
| C21 | S20 | INIL03g15019 | XM_019334098.1 | 3 | 20,121,738 | 20,123,288 | + | 270 | 3 |
| C21 | S20 | INIL08g31042 | XM_019304544.1 | 8 | 6,850,033 | 6,851,819 | - | 161 | 4 |
| C21 | S20 | INIL09g33277 | XM_019307621.1 | 9 | 21,635,093 | 21,637,187 | + | 256 | 2 |

(Continued)

**Table 1.** (Continued)

| Subfamily | | Gene ID | Gnomon (NCBI) | Chromosome number | Position on chromosome | | strand | Number of amino acids | Number of exons |
|---|---|---|---|---|---|---|---|---|---|
| Morning glory | Arabidopsis | | | | start | end | | | |
| C22 | S19 | INIL13g07867 | XM_019324507.1 | 13 | 3,970,381 | 3,972,335 | - | 276 | 4 |
| C22 | S19 | INIL05g32166 | XM_019305811.1 | 5 | 6,487,045 | 6,489,795 | + | 202 | 3 |
| C22 | S19 | INIL11g09839 | XM_019327880.1 | 11 | 1,630,643 | 1,632,375 | + | 209 | 3 |
| C23 | S18 | INIL03g17808 | XM_019338754.1 | 3 | 34,489,427 | 34491554 | - | 488 | 3 |
| C23 | S18 | INIL07g33338 | XM_019308404.1 XM_019308405.1 XM_019308406.1 | 7 | 17,374,648 | 17,378,922 | + | 555 | 3 |
| C23 | S18 | INIL08g20855 | XM_019342298.1 XM_019342299.1 XM_019342300.1 | 8 | 768,427 | 774,261 | - | 485 | 3 |
| C23 | S18 | INIL12g21835 | XM_019343649.1 | 12 | 56,783,326 | 56,787,720 | - | 546 | 2 |
| C24 | | INIL04g31722 | XM_019305531.1 | 4 | 25,016,832 | 25,018,832 | - | 316 | 3 |
| C24 | | INIL12g01339 | XM_019336624.1 | 12 | 1,998,415 | 2,000,547 | - | 356 | 3 |
| C25 | | INIL08g38640 | XM_019315175.1 XM_019315176.1 XM_019315178.1 | 8 | 39,072,851 | 39,076,646 | + | 479 | 12 |
| C25 | | INIL15g23810 | XM_019295130.1 XM_019295131.1 | 15 | 19,600,746 | 19,605,217 | - | 435 | 12 |
| C26 | S25 | INIL07g06211 | XM_019322785.1 | 7 | 5,237,655 | 5,239,775 | + | 436 | 4 |
| C26 | S25 | INIL07g06212 | XM_019322786.1 | 7 | 5,282,005 | 5,283,994 | + | 437 | 4 |
| C26 | S25 | INIL08g13864 | - | 8 | 12,385,280 | 12,395,420 | + | 496 | 7 |
| C26 | S25 | INIL10g13265 | XM_019331006.1 | 10 | 10,852,867 | 10,855,271 | + | 400 | 3 |
| C26 | S25 | INIL10g42763 | XM_019318368.1 | 10 | 26,456,385 | 26,461,974 | + | 442 | 4 |
| C26 | S25 | INIL11g18427 | "XM_019337230.1 XM_019337231.1" | 11 | 23,717,996 | 23,723,540 | + | 1000 | 4 |
| C26 | S25 | INIL12g03514 | - | 12 | 62,534,852 | 62,537,936 | + | 376 | 3 |
| C26 | S25 | INIL00g27132 | XM_019299745.1 | scaffold2396 | 41,996 | 43,987 | + | 437 | 4 |
| C26 | S25 | INIL00g27134 | XM_019299747.1 | scaffold2396 | 81,536 | 83,702 | + | 426 | 5 |
| C27 | S23 | INIL05g24219 | XM_019296142.1 | 5 | 39,630,059 | 39,634,323 | + | 413 | 2 |
| C27 | S23 | INIL05g31792 | XM_019305543.1 | 5 | 11,790,376 | 11,794,185 | - | 418 | 2 |
| C28 | S22 | INIL01g25379 | XM_019297609.1 XM_019297610.1 | 1 | 40,455,242 | 40,456,541 | + | 310 | 1 |
| C28 | S22 | INIL01g25431 | XM_019296795.1 XM_019296796.1 | 1 | 40,067,944 | 40,068,983 | - | 248 | 1 |
| C28 | S22 | INIL02g17023 | XM_019336024.1 XM_019336222.1 | 2 | 41,751,893 | 41,759,347 | + | 490 | 4 |
| C28 | S22 | INIL06g38304 | XM_019313214.1 | 6 | 1,088,480 | 1,089,498 | + | 284 | 1 |
| C28 | S22 | INIL10g12856 | XM_019330943.1 | 10 | 6,140,664 | 6,141,561 | - | 231 | 2 |
| C28 | S22 | INIL14g41452 | XM_019318549.1 | 14 | 58,438,380 | 58,439,332 | + | 269 | 1 |
| C28 | S22 | INIL14g41510 | XM_019319149.1 | 14 | 58,016,155 | 58,017,432 | - | 297 | 1 |
| C28 | S22 | INIL15g27998 | XM_019301166.1 | 15 | 3,986,197 | 3987766 | + | 348 | 1 |
| C28 | S22 | INIL15g29270 | XM_019302518.1 | 15 | 25,282,729 | 25,284,526 | + | 344 | 2 |
| C29 | S21 | INIL08g38617 | XM_019315081.1 XM_019315082.1 | 8 | 38,822,489 | 38,825,155 | + | 355 | 5 |
| C29 | S21 | INIL09g36145 | XM_019311794.1 | 9 | 10,412,907 | 10,416,698 | + | 252 | 3 |
| C29 | S21 | INIL10g12608 | XM_019330829.1 | 10 | 3,850,307 | 3,852,298 | - | 359 | 5 |
| C29 | S21 | INIL14g35326 | XM_019310126.1 | 14 | 50,677,228 | 50,678,541 | - | 264 | 2 |
| C29 | S21 | INIL14g41878 | XM_019318795.1 | 14 | 54,482,443 | 54,484,119 | + | 269 | 3 |

(Continued)

**Table 1.** (Continued)

| Subfamily | | Gene ID | Gnomon (NCBI) | Chromosome number | Position on chromosome | | strand | Number of amino acids | Number of exons |
| Morning glory | Arabidopsis | | | | start | end | | | |
|---|---|---|---|---|---|---|---|---|---|
| | | INIL01g36685 | XM_019312401.1 | 1 | 4,750,136 | 4,751,377 | - | 230 | 4 |
| | | INIL02g16845 | "XM_019336280.1 XM_019336291.1" | 2 | 40,440,393 | 40447357 | + | 530 | 13 |
| | | INIL02g17103 | XM_019336309.1 | 2 | 42,313,284 | 42,314,620 | - | 201 | 3 |
| | | INIL04g09009 | XM_019326453.1 | 4 | 41,087,133 | 41,089,320 | - | 331 | 2 |
| | | INIL04g32702 | XM_019307395.1 XM_019307396.1 XM_019307397.1 XM_019307398.1 | 4 | 5,726,191 | 5,729,159 | - | 359 | 1 |
| | | INIL05g22742 | XM_019294163.1 | 5 | 31,169,266 | 31,171,433 | - | 416 | 3 |
| | | INIL11g18974 | XM_019339442.1 XM_019339443.1 XM_019339444.1 XM_019339445.1 | 11 | 5,940,454 | 5,942,722 | - | 269 | 3 |
| | | INIL12g01151 | XM_019331043.1 | 12 | 3,476,626 | 3,479,529 | + | 308 | 3 |
| | | INIL12g03255 | XM_019306706.1 | 12 | 64,661,271 | 64,662,685 | - | 240 | 3 |
| | | INIL13g15530 | XM_019335288.1 | 13 | 12,733,089 | 12,736,028 | + | 289 | 2 |
| | | INIL05g14256 | XM_019333608.1 | 15 | 11,477,909 | 11485736 | - | 249 | 3 |

respectively. Motif 1 straddles helix 3 in the R2 repeat and helix 1 in the R3 repeat. Motif 4 and 2 composes helices 2 and 3 in the R3 repeat, respectively. In addition to the MYB domains, InR2R3-MYBs have subfamily-specific protein motifs, such as motif 7 and 10 in C2, motif 11 and 17 in C5, motif 14 in C16, motif 15 and 18 in C18, motif 16 in C23 and motif 19 in C28. These unique motifs might be related to functional differentiation of each subfamily.

## Consensus amino acid sequence in the MYB domains of InR2R3-MYBs

The MYB domains of R2R3-MYB contain highly conserved sequences [2]. To determine the consensus amino acid sequence in the MYB domains of InR2R3-MYBs, Fig 3 shows the sequence logos of the R2 and R3 repeats in the InR2R3-MYBs.

Three regularly spaced tryptophan (W) residues in typical MYB domains are important for interaction with specific DNA sequences [8]. All InR2R3-MYBs, except INIL09g35855, had three W residues in the R2 repeat (Fig 3). In the R3 repeat, the first W residue is occasionally replaced by a hydrophobic amino acid, such as phenylalanine (F), isoleucine (I) or leucine (L), which is known for R2R3-MYBs in other plant species [11, 17]. The second W residue in the R3 repeat was conserved in all InR2R3-MYBs, whereas the third W residue was conserved in most InR2R3-MYBs but not in INIL02g17103, INIL04g09009, INIL08g38640, INIL15g23810 (replaced by F), INIL11g18427 (replaced by tyrosine (Y)), and INIL11g40874 (replaced by cysteine (C)).

Conserved amino acid residues in the MYB domains were mainly distributed between the second and third conserved W residues in both R2 and R3 repeats (Fig 3). The region between the second and third W residues corresponds to helices 2 and helices 3 and their connecting loop (helix-turn-helix), and the region is important for binding to DNA [7, 8]. In particular, the third helix of each repeat (recognition helix) is essential for the direct interaction with DNA [59]. Therefore, the third helix of the MYB domain in each repeat is highly conserved.

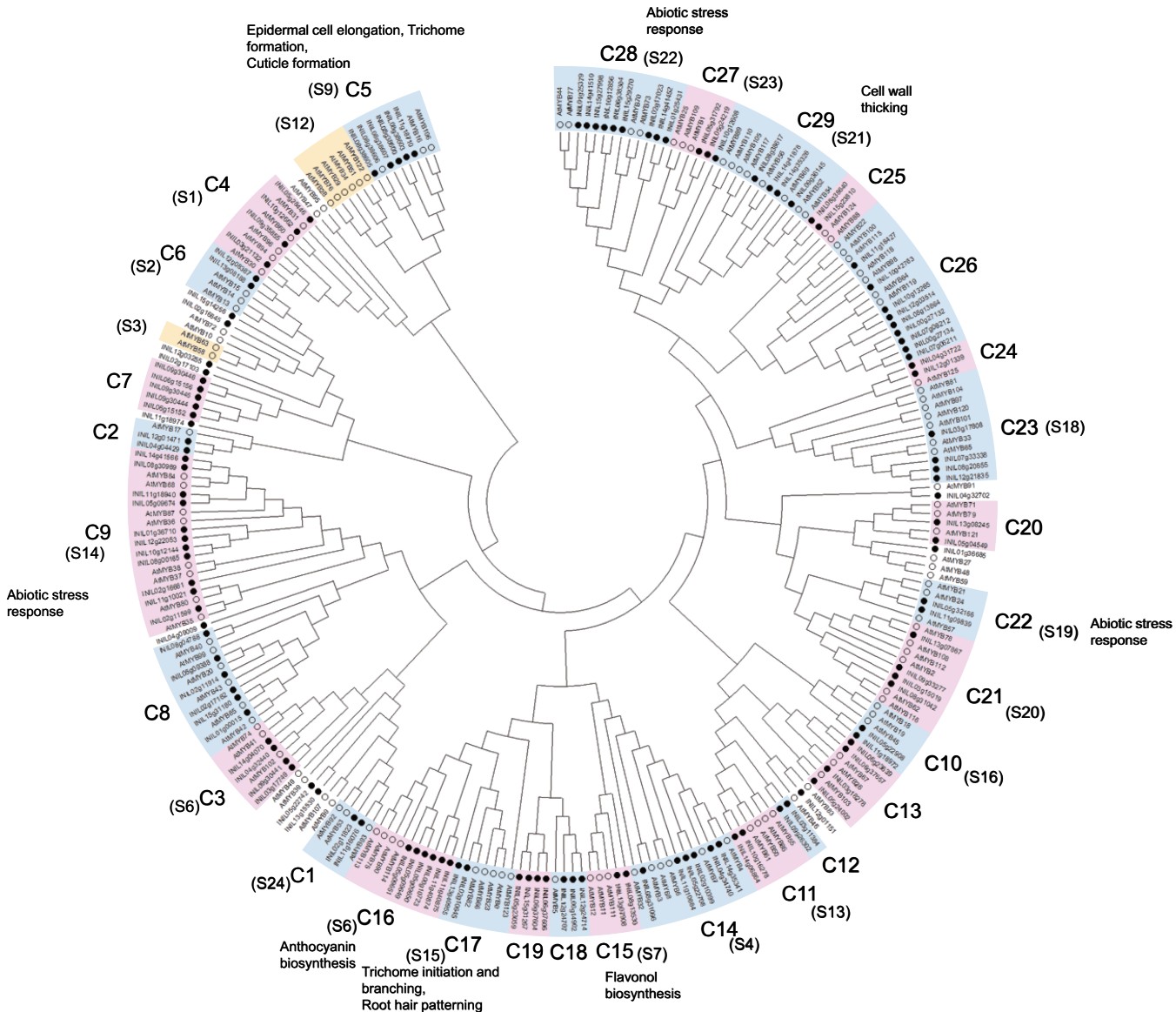

**Fig 1. Phylogenetic tree of the R2R3-MYBs of Japanese morning glory and *Arabidopsis*.** Phylogenetic tree was generated by the neighbor-joining method derived from a CLUSTAL alignment of the amino acid sequences of *Arabidopsis* [9] and Japanese morning glory R2R3-MYBs. The hollow circles represent the *Arabidopsis* R2R3-MYBs, whole the solid black circles represent the Japanese morning glory R2R3-MYBs. The functions of *Arabidopsis* R2R3-MYBs [9] were described.

## Chromosomal location of InR2R3-MYB genes

The chromosomal location of 127 *InR2R3-MYB* genes is shown in Fig 4. A total of 123 *InR2R3-MYBs* were mapped on 15 pseudo-chromosomes, while four *InR2R3-MYBs* (*INIL00g10723*, *INIL00g14902*, *INIL00g27132*, and INIL*00g27134*) were not mapped to pseudo-chromosomes but to scaffolds.

The distribution of *InR2R3-MYBs* in the pseudo-chromosomes were uneven. Chr. 8 harbored the largest number of 15 *InR2R3-MYBs*, followed by Chr. 5 with 14 *InR2R3-MYBs*. In contrast, the chromosome with the lowest number of *InR2R3-MYBs* was Chr. 7 (3 genes). Most *InR2R3-MYBs* locate at both ends of the chromosomes. In particular, relatively high

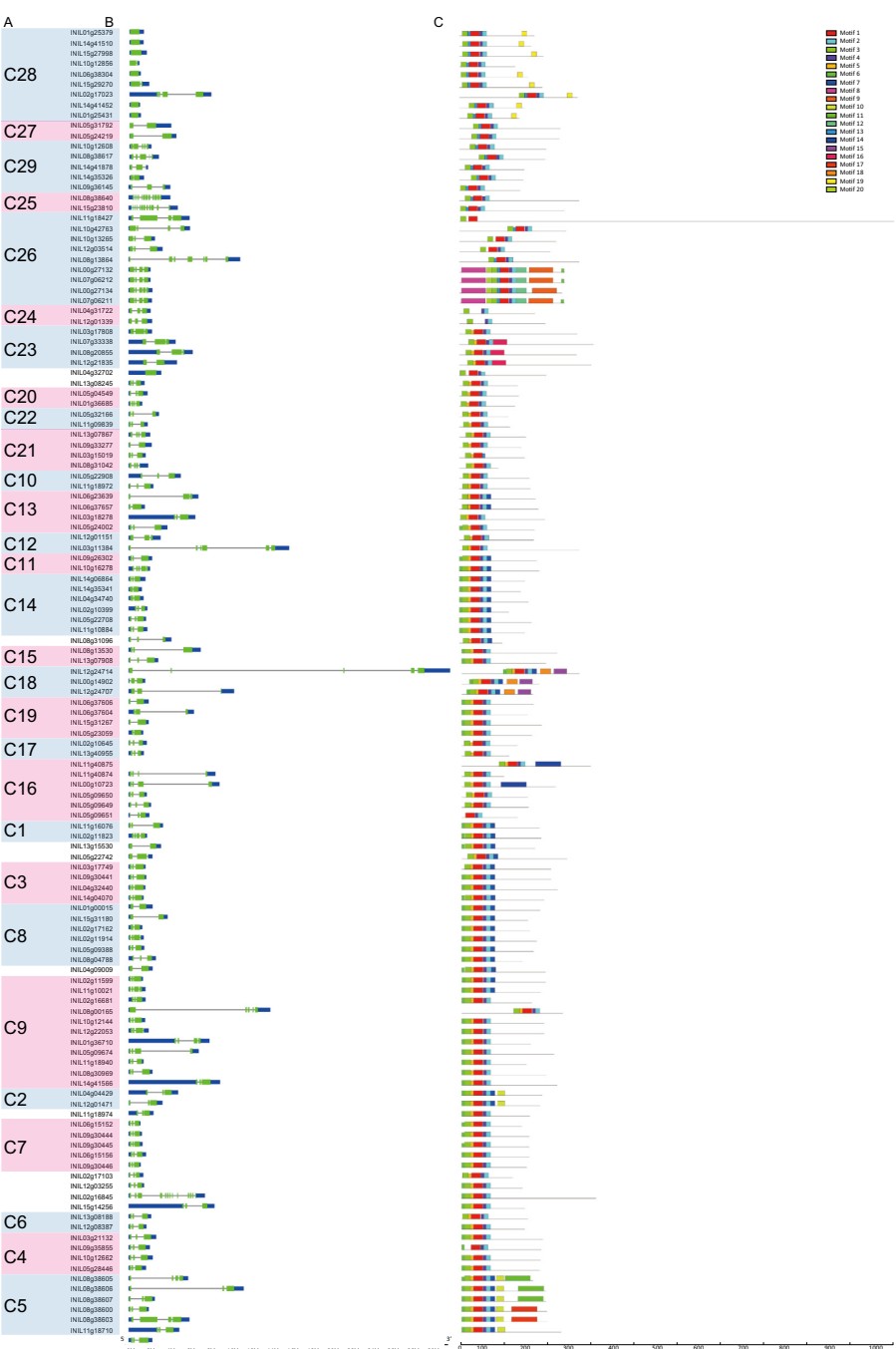

**Fig 2. Gene structures and conserved protein motifs of the Japanese morning glory R2R3-MYBs.** A: The list of R2R3-MYBs of the Japanese morning glory. The 127 R2R3-MYBs were clustered into 29 subfamilies. Numbers in parentheses indicates subfamily names of *Arabidopsis* R2R3-MYBs [9]. B: Exon-intron structures of the *R2R3-MYBs*. Exons are shown as green boxes, introns as black lines, and untranslated regions as blue boxes. C: Conversed protein motifs of the Japanese morning glory R2R3-MYBs. Motifs were identified using the MEME web server (https://meme-suite.org/meme/tools/meme). Different motifs are represented by different colored boxes. Details of the motifs were described in S3 Table.

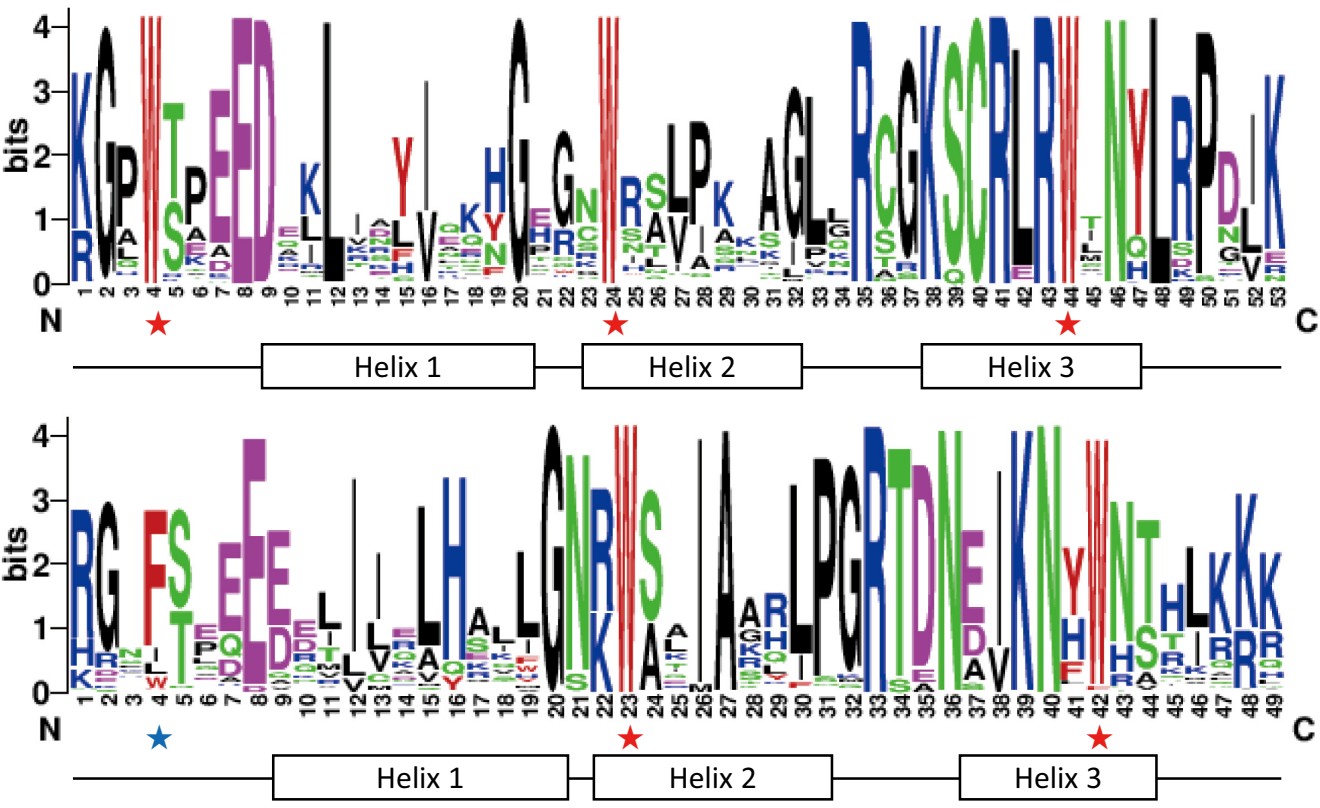

**Fig 3. Sequence logos of the R2 and R3 domains of the Japanese morning glory R2R3-MYBs.** Sequence logos were generated using WebLogo (http://weblogo. berkeley.edu/logo.cgi). Alignment analysis of R2 and R3 domains was performed with ClustalW and manually optimized. The overall height of each stack indicates the conservation of the amino acid residue at that position, and the bit score exhibits the relative frequency of the corresponding amino acid residue. The conserved tryptophan residues (W) are marked with red stars, while replaced residue in the R3 domain is marked with the blue star.

densities of *R2R3-MYBs* were observed in both arms of Chr. 5 and Chr. 8. Most central regions of these chromosomes lacked *InR2R3-MYBs*. These trends are consistent with those in tomato [11] and potato [17].

Tandem duplications of InR2R3-MYBs in the morning glory genome were estimated following the method of Huang et al. (2012) [60], that is, two or more homologous genes in a 100 kb chromosome region were defined as tandem duplicated genes. As shown in Fig 4, seven clusters of the tandem duplicated *InR2R3-MYBs* were identified as follows: threegenes in Chr. 5 *(INIL05g09649, INIL05g09650, INIL05g09651)*, two genes on Chr. 6 *(INIL06g37606, INIL06g37604)*, two genes on Chr. 6 *(INIL06g15152, INIL06g15156)*, two genes on Chr. 7 *(INIL07g06211, INIL07g06212)*, five genes on Chr. 8 *(INIL08g38600, INIL08g38603, INIL08g38605, INIL08g38606, INIL08g38607)*, three genes on Chr. 9 *(INIL09g30444, INIL09g30445, INIL09g30446)* and two genes on Chr. 11 *(INIL11g40874, INIL11g40875)*. These *InR2R3-MYBs* are thought to be the result of gene duplication.

## Functions of InR2R3-MYBs

In general, paralogs and orthologues have similar functions, and subfamily members are likely to share a common evolutionary origin and similar functions. Therefore, the functions of the morning glory InR2R3-MYB belonging to the 29 subfamilies (C1–29) were estimated based on the known functions of *Arabidopsis* AtR2R3-MYBs in the 23 subfamilies (S1–S25) (Fig 1).

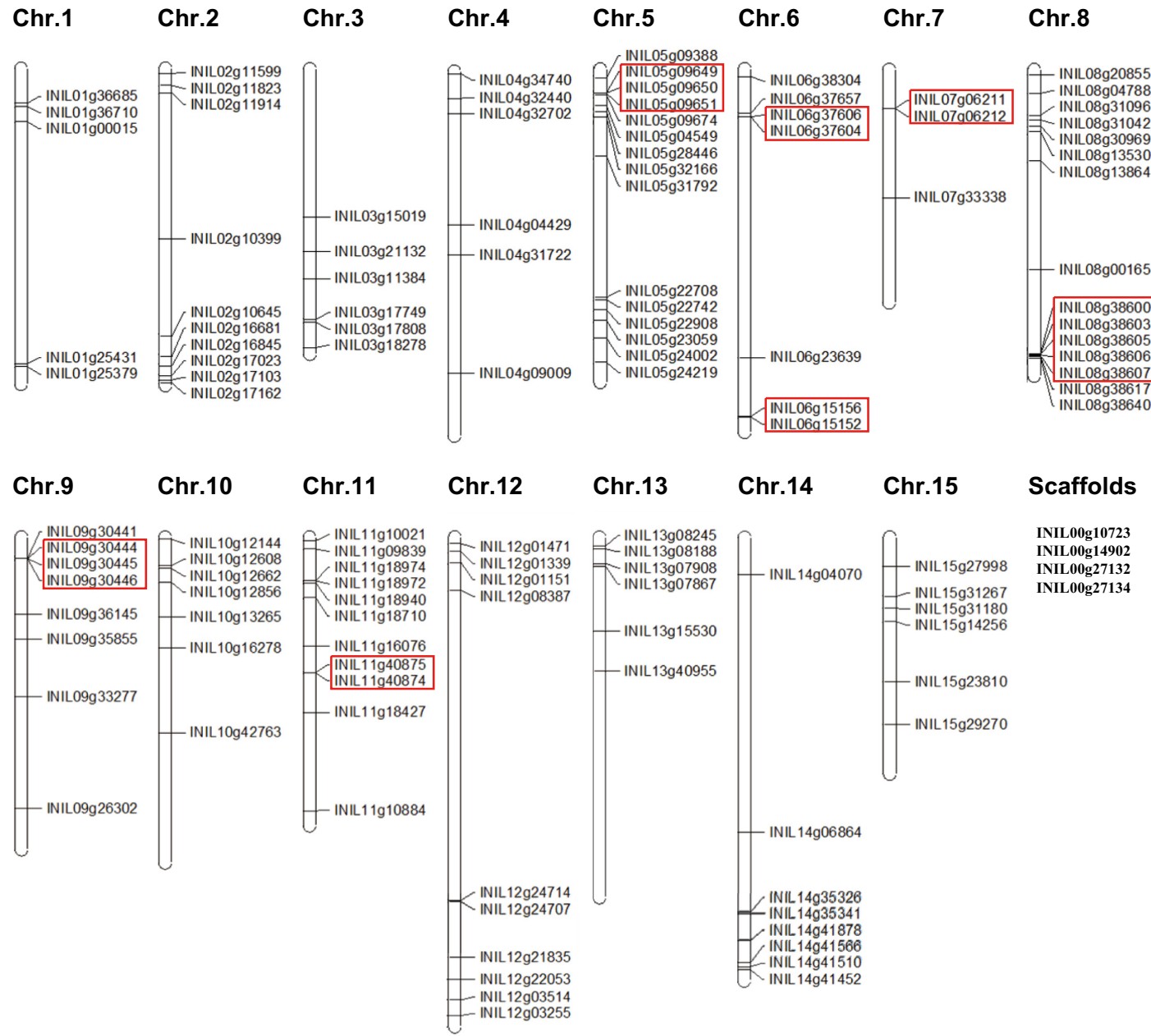

**Fig 4. Chromosomal locations of *R2R3-MYB* genes of Japanese morning glory.** The chromosomal positions of the Japanese morning glory *R2R3-MYB* gene were mapped according to the Asagao Genome Database. A total of 123 of the *R2R3-MYB* genes are mapped on the 15 chromosomes, and four genes are mapped on the scaffolds. Red boxes indicate duplicated gene clusters on the chromosomes.

C5 of morning glory corresponds to S9 of *Arabidopsis*, which includes AtMYB16 and AtMYB106 (NOK). AtMYB16 and AtMYB106 are MIXTA-like proteins that regulate petal and leaf epidermal cell elongation, trichome formation, and cuticle formation [28, 61–63]. Therefore, six InR2R3-MYBs in C5 may be involved in the regulation of petal and leaf epidermal cell elongation, trichome formation, and cuticle formation in morning glory.

Further, C9 of morning glory corresponds to S14 of *Arabidopsis*, which includes AtMYB37 (RAX2) and AtMYB84 (RAX3). These genes regulate lateral organ formation [64, 65]. Therefore, members of C9 may be involved in the regulation of cell division, such as the development of lateral organ formation in Japanese morning glory.

C15 of morning glory corresponds to S7 of *Arabidopsis*, which includes AtMYB11 (PFG2), AtMYB12 (PFG1) and AtMYB111 (PFG3). Because AtMYB11, AtMYB12 and AtMYB111 regulate flavonol biosynthesis [19], members of C15 may be involved in the regulation of flavonol biosynthesis in Japanese morning glory.

Additionally, C16 of morning glory corresponds to S6 of *Arabidopsis*, which includes AtMYB75 (PAP1) and AtMYB90 (PAP2). PAP1 and PAP2 regulate anthocyanin biosynthesis [18]. SixInR2R3-MYBs present in C16 may regulate anthocyanin biosynthesis in the morning glory. The function of C16 is discussed in detail in the following section.

C28 of the morning glory corresponds to S22 of *Arabidopsis*, which includes AtMYB44, AtMYB70 and AtMYB73. The expression of these genes is induced by abiotic stresses, such as drought and wounding [66]. Therefore, the InR2R3-MYBs in C28 may be involved in the regulation of abiotic stress responses.

## Gene expression and physiological functions of InR2R3-MYBs

To understand the organ-specific gene expression patterns of *InR2R3-MYB*s, the RNAseq data of *InR2R3-MYB*s in six tissues (embryo, flower, leaf, root, seed coat, and stem) were obtained from the Asagao Genome Database (http://viewer.shigen.info/asagao/). The data were projected on a heat map, as shown in Fig 5.

Relatively high gene expression (RPKM>5) in all analyzed organs were observed in *INIL04g32702*, *INIL11g18427* and *INIL15g27998* (S5 Table). *INIL04g32702* was homologous to *AtMYB91* (*AS1*), which regulates leaf morphogenesis in *Arabidopsis* [67]. *INIL04g32702* expression was high in embryos, flowers and leaves; thus, *INIL04g32702* may be involved in their morphogenesis.

The expression of 60 *InR2R3-MYB*s was low in the all analyzed organs (RPKM<2), and no expression was observed for eight genes (*INIL06g37606*, *INIL08g20855*, *INIL10g42763*, *INIL12g03514*, *INIL12g22053*, *INIL12g24714*, *INIL13g07867*, and *INIL15g29270*) in all organs (S5 Table).

A number of *InR2R3-MYB*s showed organ-specific relatively high gene expression levels (RPKM>5). Eight *InR2R3-MYB*s (*INIL02g11599*, *INIL11g09839*, *INIL11g18974*, *INIL11g40875*, *INIL12g01471*, *INIL13g40955*, *INIL14g04070*, and *INIL00g10723*) were highly expressed in flower. *INIL02g16845* was highly expressed specifically in leaf. Additionally, eight *InR2R3-MYB*s (*INIL02g11914*, *INIL04g32440*, *INIL05g04549*, *INIL05g09388*, *INIL05g09649*, *INIL09g30444*, *INIL09g30446*, *INIL10g12144*) and 2 *InR2R3-MYB*s (*INIL02g10645*, *INIL05g09650*) were highly expressed specifically in root or stem, respectively (S5 Table).

*INIL02g11599* and *INIL11g09839*, which showed high and specific expression in flower, were in C22 and have high homology to *Arabidopsis AtMYB35* (*TDF1*) or *AtMYB21/24*, respectively. AtMYB35 functions in the development and differentiation of tapetum tissue in anther [68]; therefore, INIL02g11599 may be involved in tapetum development. AtMYB21/24 regulates stamen filament development [69]; therefore INIL11g09839 is expected to be involved in stamen filament development. INIL12g01471, which was in the same clade with AtMYB17, was highly expressed in flower. AtMYB17 has been reported to regulate early inflorescence development [70]; therefore, INIL12g01471 may regulate early inflorescence development. INIL02g11914 and INIL05g09388, which have high homology to AtMYB20, showed high gene expression specifically in root. AtMYB20 negatively regulates drought stress

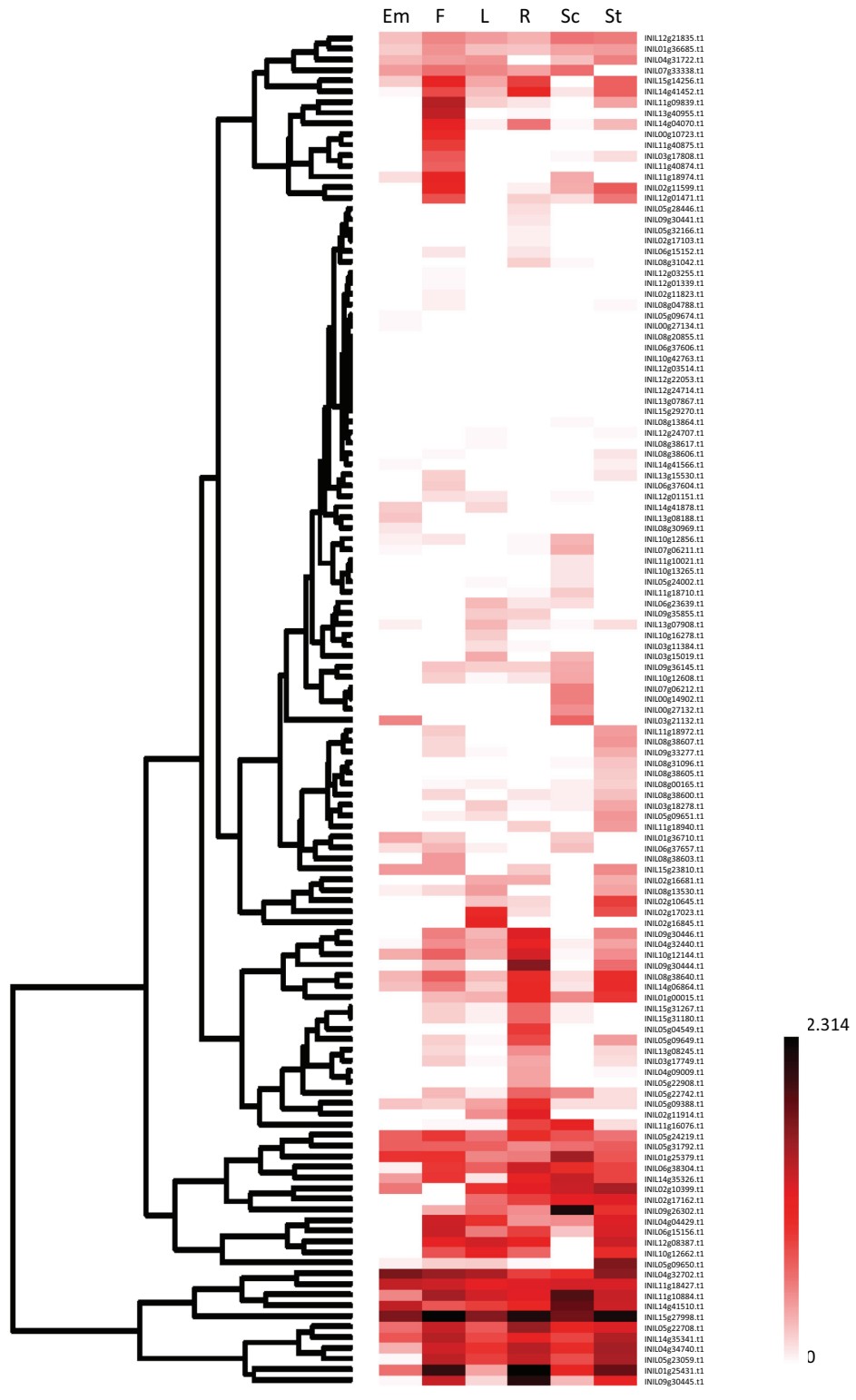

**Fig 5. Gene expression profile of the *R2R3-MYBs* in Japanese morning glory.** Gene expression data for various organs of Japanese morning glory was obtained from *Ipomoea nil* RNA-seq database (http://viewer.shigen.info/asagao/jbrowse.php?data=data/Asagao_1.2). The heat map were generated for the base 10 logarithms of the number of each RPKM value plus 1.0. Gene expression levels (low to high) are indicated by light to deep red color shades. Em: Embryo, F: Flower, L: Leaf, R: Root, Sc: Seed coat, St: Stem.

response [71] and salt stress response [72], suggesting that INIL02g11914 and INIL05g09388 may regulate abiotic stress responses.

## InR2R3-MYBs involves in anthocyanin biosynthesis

A well-known function of plant R2R3-MYBs is the regulation of anthocyanin biosynthesis, which is important in ornamental plants, including morning glory. In Japanese morning glory, InMYB1 have been reported to be involved in anthocyanin biosynthesis [42]. In addition, *InMYB2* and *InMYB3* have been reported as orthologs of petunia *AN2*, which regulates anthocyanin biosynthesis in petunia [42].

*INIL05g09650* has a predicted transcript sequence matches the cDNA sequence of *InMYB2*and thus identicalto *InMYB2*. *InMYB2* is expressed in all tissues colored with anthocyanins other than petal [42]. According to the RNA-seq database, *INIL05g09650* is expressed mostly in stems, which accumulate anthocyanins other than petals (Fig 5). This suggests that *InMYB2* and *INIL05g09650* are identical. *INIL05g09649*, which has high homology to *INIL05g09650*, was highly expressed in stems (Fig 5). Therefore, INIL05g09649 may be involved in the regulation of anthocyanin biosynthesis in the stem along with INIL05g09650.

*INIL05g09651* has the highest homology to *InMYB3*. INIL05g09651 lacks R2 repeat and contains only R2 repeat in the TKS line used in the genome database of the Japanese morning glory, while Morita et al. (2006) [42] reported that InMYB3 in KK/ZSK-2 line contains both R2 and R3 repeats. *INIL05g09651* of the TSK line has a stop codon after the region encoding the R2 repeat. This is considered an interspecific polymorphism (single nucleotide substitution), and InMYB3 (INIL05g09651) is considered to lose function in the TKS line.

*InMYB1* is expressed specifically in petal and is involved in the regulation of petal coloration (anthocyanin accumulation) in morning glory [42]. The promoter of *InMYB1* can be used as a petal-specific promoter [73–75]. *INIL00g10723*, which was not mapped to any pseudochromosome, but to a scaffold (Fig 4), has the highest homology to *InMYB1*. The upstream sequence of *INIL00g10723* was identical to the promoter region of *InMYB1*. Therefore, INIL00g10723 and InMYB1 were considered to be identical. However, the amino acid sequences of C-terminus of INIL00g10723 and InMYB1 were not identical, and an additional sequence was present in INIL00g10723. Morita et al. (2006) [42] reported that *InMYB1* has three exons, while four exons are predicted in *INIL00g10723*, and the additional sequence corresponds to exon 4. Thus, we checked the genomic sequence and RNA-seq data of *INIL00g10723* on Japanese morning glory database, and found an identical sequence to the three exons of *InMYB1*, with a stop codon after exon 3 of *INIL00g10723* (S2A and S2B Fig). Therefore, we concluded that the predicted coding sequence of *INIL00g10723* was incorrect, and InMYB1 and INIL00g10723 are identical.

Both *INIL11g40874* and *INIL11g40875*, which have high homology to *INIL00g10723*, showed petal-specific expression. The numbers of exons of these two genes differed from other C23 genes. Therefore, as with *INIL00g10723*, we checked the genomic sequences of *INIL11g40874* and *INIL11g40875*, and corrected the predicted coding regions, the transcription start points, and stop codon positions (S2 Fig). Consequently, the amino acid sequences of INIL11g40875 matched perfectly with those of InMYB1 (INIL00g10723), although the promoter region of *INIL11g40875* matched with only 212 bp upstream region of *InMYB1* and further upstream regions were not identical (S1 Fig). We concluded that *INIL11g40875* and *InMYB1* (*INIL00g10723*) are different genes that are thought to be produced by gene duplication.

The genomic sequence, including upstream and downstream regions, of *INIL11g40874* is identical to that *InMYB1* (*INIL00g10723*), except for exon 3 and its downstream. The nonidentical region corresponds to the linkage point of contigs and the sequence is considered to

be erroneous. Therefore, we *INIL00g10723*, *INIL11g40874* and *InMYB1* may be identical gene on Chr. 11. Our final discussion of C16 is summarized in S3 Fig.

## Conclusion

In this study, we performed genome-wide analysis of R2R3-MYB transcription factors in Japanese morning glory. A total of 126 *InR2R3-MYBs* were identified in the Japanese morning glory genome and their information, including gene structures, protein motifs and gene expression profiles, was collected. Our phylogenetic tree analysis revealed the presence of 29 subfamilies of InR2R3-MYBs, and the predicted functions of each subfamily have been discussed using gene expression profile and based on the functions of *Arabidopsis AtR2R3-MYBs*. This study provides essential and useful information for further functional and physiological studies on InR2R3-MYBs in morning glory.

## Supporting information

**S1 Fig. Sequence alignment of the 5' upstream regions of INIL00g17023, INIL11g40874 and INIL11g40875.** The 1026-bp upstream sequences of INIL00g17023, INIL11g40874 and INIL11g40875 from the transcription start site were aligned. The number above the alignment indicates the position from the transcription start site.
(TIF)

**S2 Fig. Gene structures and genomic sequences of *INIL00g17023*, *INIL11g40874* and *INIL11g40875*.** A: *INIL00g10723* and *INIL11g40875* have stop codons in the same position as InMYB1, suggesting that they have three exons, as with *InMYB1*. Although the sequence of this region in *INIL11g40874* is unknown because it corresponds to the linkage of contigs, the high homology of the other parts of the sequence suggests that it has three exons as well. B: RNA-seq data of *INIL00g10723* supported that InMYB1 contain three exons, not four.
(TIF)

**S3 Fig. Phylogenetic tree of the C16.** Phylogenetic tree was generated by the neighbor-joining method derived from a CLUSTAL alignment of the amino acid sequences of six members of C16.
(TIF)

**S1 Table. Amino acid sequences of 127 InR2R3-MYBs.**
(DOCX)

**S2 Table. The list of the 1R-MYBs, 3R-MYBs and 4R-MYB identified in the genome of Japanese morning glory.**
(XLSX)

**S3 Table. Sequences of the conserved motifs among the Japanese morning glory R2R3-MYBs.**
(XLSX)

**S4 Table. Insertion and deletion of amino acid residue in the R2 and R3 domains of the Japanese morning glory R2R3-MYBs.**
(XLSX)

**S5 Table. RPKM value of RNA-seq of the Japanese morning glory the Japanese morning glory R2R3-MYBs.**
(XLSX)

## Acknowledgments

We thank the National Bioresource Project (NBRP) Moring glory for the discussion about the genome data and the reconfirmation of the annotations.

## Author Contributions

**Conceptualization:** Katsuhiro Shiratake.

**Data curation:** Atsushi Hoshino, Shungo Otagaki.

**Funding acquisition:** Katsuhiro Shiratake.

**Investigation:** Ayane Komatsuzaki.

**Project administration:** Katsuhiro Shiratake.

**Resources:** Atsushi Hoshino.

**Supervision:** Katsuhiro Shiratake.

**Writing – original draft:** Ayane Komatsuzaki, Katsuhiro Shiratake.

**Writing – review & editing:** Ayane Komatsuzaki, Atsushi Hoshino, Shungo Otagaki, Shogo Matsumoto, Katsuhiro Shiratake.

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
