## [Decision Letter · Decision Letter 0]

25 Jul 2022

PONE-D-22-17708Genome-wide analysis of R2R3-MYB transcription factors in Japanese morning gloryPLOS ONE

Dear Dr. Komatsuzaki,

Thank you for submitting your manuscript to PLOS ONE. After careful consideration, we feel that it has merit but does not fully meet PLOS ONE’s publication criteria as it currently stands. Therefore, we invite you to submit a revised version of the manuscript that addresses the points raised during the review process.

We look forward to receiving your revised manuscript.

Kind regards,

Hiroshi Ezura

Academic Editor

PLOS ONE

Journal Requirements:

"We thank the National Bioresource Project (NBRP) Moring glory for the discussion about the genome data and the reconfirmation of the annotations. This work was supported partially by the Grant-in-Aids for Scientific Research (KAKENHI: 15H04449, 16K14850, 20K20372, 18H03950, 21K19111, 21H02184) from the Japan Society for the Promotion of Science (JSPS)."

"This work was supported partially by the Grant-in-Aids for Scientific Research (KAKENHI: 15H04449, 16K14850, 20K20372, 18H03950, 21K19111, 21H02184) to K.S. from the Japan Society for the Promotion of Science (JSPS: https://www.jsps.go.jp/english/index.html). The funders had no role in study design, data collection and analysis, decision to publish, or preparation of the manuscript."

Reviewers' comments:

Reviewer's Responses to Questions

**Comments to the Author**

1. Is the manuscript technically sound, and do the data support the conclusions?

Reviewer #1: Yes

Reviewer #2: Yes

2. Has the statistical analysis been performed appropriately and rigorously? 

Reviewer #1: Yes

Reviewer #2: I Don't Know

3. Have the authors made all data underlying the findings in their manuscript fully available?

Reviewer #1: Yes

Reviewer #2: Yes

4. Is the manuscript presented in an intelligible fashion and written in standard English?

Reviewer #1: Yes

Reviewer #2: Yes

5. Review Comments to the Author

Reviewer #1: In this study, Komatsuzaki et al. identified 126 R2R3-MYB genes by genome-wide analysis in

Japanese morning glory. This study will provide useful information for studies on MYB transcription factors.

The authors should provide more detailed information about the samples used in the gene expression analysis, such as stages, growth conditions, number of replicates.

Reviewer #2: This paper describes a genome-wide analysis of R2R3-MYB transcription factors in Japanese morning glory. The manuscript is well-written and this paper will supply basic information to researchers on MYB transcription factors and related areas. However, this paper contains few but crucial problems.

Clade 16:

Because the C16 seems to be one the most extensively studied R2R3-MYB class in this plant, the definitive results should be described with new supplemental figures.

How many members belong to C16? It should be clearly described in abstract. In the current abstract, 3 known genes, named InMYB1, InMYB2 and InMYB3, and 3 new genes were described. Therefore, 6 genes are belonged to C16. However, in Figs. 1, 2, and 4, only 5 genes were presented. In Fig. 5, I could find only 3 genes. This problem contains not only for the gene number but also for the gene member.

I think InMYB3 should be add to the C16 members, I agree the InMYB3 is identical to INIL05g09651. The authors described the existence of the interspecific polymorphism in INIL05g09651, that cause wrong annotation in the database the authors used. However, InMYB3 was already published and those results should be respected. The authors must present the polymorphism with nucleotide sequences in a new supplemental figure, and modified all the figures and tables about INIL05g09651. Especially in the Table S2, there are several Remarks “It could be R2R3-MYB if the annotation is corrected.” The authors should try to eliminate these sentences, as much as possible.

The authors describe just before “Conclusion”, “Therefore, we concluded that INIL00g12723, INIL11g40874 and InMYB1 are identical and present on Chr. 11”. I think this conclusion needs more data. One question is about the existence of 4th exon. If INIL1140874 and InMYB1 are identical, mRNA of InMYB1 should have 4th exon. The authors should mention about 4th exon and its downstream of INIL1140874 and InMYB1. Do the 4th exon proteins show conservation beyond species?

If the problem of InMYB1 is clearly solved, the members of C16 will be five with InMYB3.

InMYB1=INIL00g10723= INIL11g40874,

INIL11g40875,

InMYB2=INIL05g09650,

INIL05g09649.

InMYB3= INIL05g09651

This conclusion should be reflected in Abstract and all the figures and the tables.

Other improvements:

The authors should clearly describe about, how the number of clades determined and how the number of orphan genes determined. The number of orphan genes, that not belong to any clade, are not consistent, between 9 to 11 in throughout of the manuscript. In the Results and Discussions, the authors described “Ten R2R3-MYBs did not belong to any clade.” However, I found at least, in the Fig. 1, nine, in the Fig. 2, eleven, and in the Table1, eleven.

The number of clade members are not consistent in some clade.

(Table 1; Fig. 1; Fig. 2)

C6 (4; 5; 4)

C14 (5; 6; 5)

C21 (3; 4; 3)

C22 (3; 3; 2)

In Fig. 2, the numbers in parentheses, the subfamily names of Arabidopsis R2R3-MYBs should be presented more. Currently there is only (S9).

In Fig. 2, the colors of motifs are sometimes similar and difficult to distinguish them, the authors should make a high resolution figure. About these motifs, accompanied short descriptions will help the readers for better understanding. Could the authors add short descriptions for each motif in Fig. 2 or Table S3? In the Table S3, some important amino-acid residues can be designated in red, as in the Table S5, i.g., W residue. Please make difference clearer between the MYB domains and other domains.

In Table S2, there are several Remarks “It could be R2R3-MYB if the annotation is corrected.” The authors should try to eliminate these sentences, as much as possible. At least for INIL05g09651†, the problem should be solved and another remark about polymorphism can be added. There is no Remarks to INIL13g07537*.

Despite of several problems, this manuscript is essentially valuable and should be published after improvement.

6. PLOS authors have the option to publish the peer review history of their article (what does this mean?). If published, this will include your full peer review and any attached files.

Reviewer #1: No

Reviewer #2: No

---

## [Author Response · Author response to Decision Letter 0]

8 Sep 2022

Reviewer 1: In this study, Komatsuzaki et al. identified 126 R2R3-MYB genes by genome-wide analysis in Japanese morning glory. This study will provide useful information for studies on MYB transcription factors.

Response: Thank you for reviewing our manuscript and considering our manuscript as sound and intelligible scientific research. We really appreciate your advice and comments which have helped to improve the quality of our manuscript. Below are our point-by-point responses.

The authors should provide more detailed information about the samples used in the gene expression analysis, such as stages, growth conditions, number of replicates.

Response: Thank you for your suggestion. We added an explanation about the samples in Line 136-140 on Page 7.

Reviewer 2: This paper describes a genome-wide analysis of R2R3-MYB transcription factors in Japanese morning glory. The manuscript is well-written and this paper will supply basic information to researchers on MYB transcription factors and related areas. However, this paper contains few but crucial problems.

Response: Thank you for reviewing our manuscript and finding high scientific values in our manuscript. We really appreciate your advice and comments which have helped to improve the quality of our manuscript. Below are our point-by-point responses.

Clade 16:

Because the C16 seems to be one the most extensively studied R2R3-MYB class in this plant, the definitive results should be described with new supplemental figures.

Response: Thank you for your helpful suggestion. According to your suggestion, we added the summarized result (phylogenetic tree of C16) as Figure S3.

How many members belong to C16? It should be clearly described in abstract. In the current abstract, 3 known genes, named InMYB1, InMYB2 and InMYB3, and 3 new genes were described. Therefore, 6 genes are belonged to C16. However, in Figs. 1, 2, and 4, only 5 genes were presented. In Fig. 5, I could find only 3 genes. This problem contains not only for the gene number but also for the gene member.

Response: Thank you for your helpful suggestion. We revised the number of C16 members as 6, including InMYB3 (INIL05g09651), as summarized in Figure S3. And the text, Fig. 1, Fig. 2, Fig.3, Fig. 4 and Fig. 5 were revised. In Fig. 5, the genes are clustered by gene expression pattern, but not by homology. So, some members of C16 appeared at distant locations.

I think InMYB3 should be add to the C16 members, I agree the InMYB3 is identical to INIL05g09651. The authors described the existence of the interspecific polymorphism in INIL05g09651, that cause wrong annotation in the database the authors used. However, InMYB3 was already published and those results should be respected. The authors must present the polymorphism with nucleotide sequences in a new supplemental figure, and modified all the figures and tables about INIL05g09651. Especially in the Table S2, there are several Remarks “It could be R2R3-MYB if the annotation is corrected.” The authors should try to eliminate these sentences, as much as possible.

Response: Thank you for your helpful suggestion. I agreed with your suggestion. According to your suggestion, we added INIL05g09651 to the C16 members. We also revised the remarks “It could be R2R3-MYB if the annotation is corrected” in Table S2 to be as specific as possible.

The authors describe just before “Conclusion”, “Therefore, we concluded that INIL00g12723, INIL11g40874 and InMYB1 are identical and present on Chr. 11”. I think this conclusion needs more data. 

Response: Thank you for your helpful suggestion. I agreed with your suggestions. Since there was insufficient data to draw a conclusion, the wording was changed to express speculation.

One question is about the existence of 4th exon. If INIL1140874 and InMYB1 are identical, mRNA of InMYB1 should have 4th exon. The authors should mention about 4th exon and its downstream of INIL1140874 and InMYB1. Do the 4th exon proteins show conservation beyond species?

Response: Thank you for your comments. We apologize for the misunderstanding due to our poor writing. We checked the RNA-seq data of the Japanese morning glory genome database and found no sequence read was mapped on the exon 4 of INIL00g10723 and INIL11g40874, which have the highest homology to InMYB1. The mapping data was presented as Fig. S2B. Three-exon structure of InMYB1 was reported previously (Morita et al. 2006). Therefore, we concluded that the number of exons of InMYB, INIL00g10723 and INIL11g40874, are three.

If the problem of InMYB1 is clearly solved, the members of C16 will be five with InMYB3.

InMYB1=INIL00g10723= INIL11g40874,

INIL11g40875,

InMYB2=INIL05g09650,

INIL05g09649.

InMYB3= INIL05g09651

This conclusion should be reflected in Abstract and all the figures and the tables.

Response: Thank you for your helpful suggestion. There was not sufficient information to resolve the problem about InMYB1. However, we noted that InMYB1=INIL00g10723, InMYB2=INIL05g09650, and InMYB3=INIL05g09651 in Table 1 and Fig. S3. 

Other improvements:

The authors should clearly describe about, how the number of clades determined and how the number of orphan genes determined.

Response: Thank you for your helpful suggestion. We added an explanation how the number of clades was determined and how the number of orphan genes was determined in Line 116-119 on Page 6.

The number of orphan genes, that not belong to any clade, are not consistent, between 9 to 11 in throughout of the manuscript. In the Results and Discussions, the authors described “Ten R2R3-MYBs did not belong to any clade.” However, I found at least, in the Fig. 1, nine, in the Fig. 2, eleven, and in the Table1, eleven.

Response: Thank you for your pointing the mistake out. The correct number of orphan genes is 11. Ten in Results and Discussions was a mistake. Because Fig. 1 and Fig. 2A contain different members, the phylogenetic trees were slightly different, and therefore numbers of orphan genes differed. The clade was determined based on the data in Fig. 2A in the previous manuscript, but since it conflicts with Fig. 1. The clade was re-determined based on Fig. 1 and the phylogenetic tree in Fig. 2A was deleted.

The number of clade members are not consistent in some clade.

(Table 1; Fig. 1; Fig. 2)

C6 (4; 5; 4)

C14 (5; 6; 5)

C21 (3; 4; 3)

C22 (3; 3; 2)

Response: Thank you for your suggestion. Two phylogenetic trees in Fig. 1 and Fig. 2A with different members caused inconsistencies. As mentioned above, to solve this problem, we re-determined the members of clades based on Fig. 1, and the phylogenetic tree in Fig. 2A was deleted.

In Fig. 2, the numbers in parentheses, the subfamily names of Arabidopsis R2R3-MYBs should be presented more. Currently there is only (S9).

Response: As mentioned above, the phylogenetic tree in Fig. 2A was deleted.

In Fig. 2, the colors of motifs are sometimes similar and difficult to distinguish them, the authors should make a high resolution figure. 

Response: Thank you for your suggestion. However, Fig. 2 is an automatically generated by MEME, therefore it is difficult to modify.

About these motifs, accompanied short descriptions will help the readers for better understanding. Could the authors add short descriptions for each motif in Fig. 2 or Table S3? In the Table S3, some important amino-acid residues can be designated in red, as in the Table S5, i.g., W residue. Please make difference clearer between the MYB domains and other domains.

Response: Thank you for your suggestion. However, the motifs in Fig. 2 generated figure by MEME are not functional motifs, but only conserved amino acid sequences without the name.

In Table S2, there are several Remarks “It could be R2R3-MYB if the annotation is corrected.” The authors should try to eliminate these sentences, as much as possible. At least for INIL05g09651†, the problem should be solved and another remark about polymorphism can be added. There is no Remarks to INIL13g07537*.

Response: Thank you for your helpful suggestion. According to your suggestion, we revised the remarks “It could be R2R3-MYB if the annotation is corrected” in Table S2 to be as specific as possible including INIL13g07537. We decided to treat INIL05g09651 as R2R3-MYB, INIL05g09651was moved from Table S2 to Table 1.

---

## [Editor Report · Decision Letter 1]

21 Sep 2022

Genome-wide analysis of R2R3-MYB transcription factors in Japanese morning glory

PONE-D-22-17708R1

Dear Dr. Komatsuzaki,

We’re pleased to inform you that your manuscript has been judged scientifically suitable for publication and will be formally accepted for publication once it meets all outstanding technical requirements.

Kind regards,

Hiroshi Ezura

Academic Editor

PLOS ONE
---

## [Editor Report · Acceptance letter]

12 Oct 2022

PONE-D-22-17708R1 

Genome-wide analysis of R2R3-MYB transcription factors in Japanese morning glory 

Dear Dr. Komatsuzaki:

I'm pleased to inform you that your manuscript has been deemed suitable for publication in PLOS ONE. Congratulations! Your manuscript is now with our production department. 

Kind regards, 

on behalf of

Prof. Hiroshi Ezura 

Academic Editor

PLOS ONE